# Consistent, multidimensional differential histogramming and summary statistics with YODA 2

A. Buckley[1], L. Corpe[2], M. Filipovich[3], C. Gütschow[4,5,*],
N. Rozinsky[1], S. Thor[6], Y. Yeh[5], J. Yellen[1]

[1] *School of Physics & Astronomy, University of Glasgow,*
*University Place, G12 8QQ, Glasgow, UK*
[2] *LPC, Université Clermont Auvergne, CNRS/IN2P3, Clermont-Ferrand, France*
[3] *Department of Physics, University of Oxford, Clarendon Laboratory,*
*Parks Road, Oxford, OX1 3PU, UK*
[4] *Centre for Advanced Research Computing, University College London,*
*Gower Street, London, WC1E 6BT, UK*
[5] *Department of Physics & Astronomy, University College London,*
*Gower Street, WC1E 6BT, London, UK*
[6] *KTH Royal Institute of Technology, SE-100 44 Stockholm, Sweden*

November 12, 2024

## Abstract

Histogramming is often taken for granted, but the power and compactness of partially aggregated, multidimensional summary statistics, and their fundamental connection to differential and integral calculus make them formidable statistical objects, especially when very large data volumes are involved. But expressing these concepts robustly and efficiently in high-dimensional parameter spaces and for large data samples is a highly non-trivial challenge – doubly so if the resulting library is to remain usable by scientists as opposed to software engineers. In this paper we summarise the core principles required for consistent generalised histogramming, and use them to motivate the design principles and implementation mechanics of the re-engineered YODA histogramming library, a key component of physics data–model comparison and statistical interpretation in collider physics.

# 1 Introduction

In the current era of advanced statistical analysis methods and toolkits – from Bayesian inference to machine-learning – the simple concept of a histogram is often taken for granted. Yet it is worth reflecting on how powerful a tool they are: a set of summary statistics grouped into binned ranges of an independent variable or variables, with a fixed data size and a mathematical definition directly linked to differential and integral calculus. There is surprising conceptual depth in these apparently simple objects.

The first of these features means histograms are an extremely memory-efficient approach to approximating distributions. Unlike "ntuple" datasets, histogram objects are the same size regardless of how many "fill" events are aggregated into them. As CPU capability has grown much faster than RAM, this is an increasingly rather than decreasingly important feature. The mathematical foundations furthermore mean that the aggregation can be partitioned, allowing parallel operation and/or continuous updating: particularly useful and important features for processing of datasets much larger than can be held in memory. Such applications abound in the modern era, from event analysis at particle-physics facilities to analysis of datasets from hundreds of millions of digital-service users.

Unfortunately, few computational libraries implementing histograms for statistical analysis make full use of these parallels, which limits functionality and can even lead to numerical approximations [1]. The YODA statistical analysis library is an outlier in this, having been designed around a mixture of mathematical consistency and a user-friendly programmatic interface. However, in the decade since the release of YODA v1.0, limitations in that design also became apparent, leading to a ground-up redesign to generalise the ideas of the first version while respecting real-world requirements. This paper first introduces the mathematical foundations of general statistical histogramming, the resulting top-down requirements and motivations in library design, then the specific approaches taken to achieve them in (primarily) modern C++ code and supporting tools.

# 2 Statistical preliminaries

The ubiquity of histograms means it is typical to assume understanding of what they are, which can lead to issues of mismatch between design and usage. The most common such issue is a conflation between the statistical content of a histogram bin and its rendering (e.g. as bin height) in a plotted representation. To avoid such ambiguities, we first define the statistical objects of interest, before discussing the software design constraints and the technical implementation.

## 2.1 Unweighted mean and variance

We start with the first and second-order *statistical moments*, i.e. the mean and variance, of an unbinned scalar quantity $x$, as obtained from the probability density function (pdf) $f(x) \equiv dP/dx$:

$$\langle x \rangle \equiv \int_{x \in X} x f(x)\, dx \tag{1}$$

$$\langle x^2 \rangle \equiv \int_{x \in X} x^2 f(x)\, dx \tag{2}$$

$$\sigma^2(x) \equiv \langle x^2 \rangle - \langle x \rangle^2. \tag{3}$$

Of course these familiar quantities capture the average position and the typical (squared) dispersion of the $f(x)$ distribution, with perfect differential information about the density function. In reality we are typically limited in our knowledge about such such distributions as they are known not via an analytic form amenable to integration, but via a finite-sized *sample* from the distribution (typically obtained either by experiments on natural systems or by some computer code sampling the implicit distribution, e.g. a Monte Carlo event generator). From the finite sample we can build estimators to these quantities, computed over a sampled set of $N$ *fill values* $\{x_n\}$ for $1 \le n \le N$, and unbiased in the sense that they converge to the ideal values in the limit of infinite sample size, $N \to \infty$:

$$\langle \hat{x} \rangle_{\mathrm{U}} \equiv \frac{\sum_{n=1}^{N} x_n}{N} \tag{4}$$

$$\begin{aligned}
\hat{\sigma}_{\mathrm{U}}^2(\hat{x}) &\equiv \frac{\sum_{n=1}^{N}(x_n - \langle x \rangle)^2}{N-1} \\
&= \langle x^2 \rangle_{\mathrm{U}} - \langle x \rangle_{\mathrm{U}}^2 \\
&= \frac{\sum_{n=1}^{N} x_n^2}{N-1} - \frac{\left(\sum_{n=1}^{N} x_n\right)^2}{(N-1)^2} \, .
\end{aligned} \tag{5}$$

Here the $N-1$ term in the variance is the Bessel correction to achieve an unbiased estimator, one degree of freedom having already been used to obtain the estimated mean.

## 2.2 Estimated counts and efficiencies

A closely related quantity is the Poisson estimator of the uncertainty on the *count* of a random variable, i.e. the number of observed fills in a data-sampling run of fixed length. The variance of a Poisson random variable is equal to its mean, and as our best estimate of the Poisson mean is the final count, the best estimate of the Poisson variance is also the count: $\sigma^2(N)_{\mathrm{P}} = N$. As a result, the relative uncertainty on the count decreases with the classic Monte Carlo scaling, $\sigma(N)/N = 1/\sqrt{N}$.

Poisson rate is distinct from the *efficiency* of a fill, which is the fraction $\epsilon$ of a known number of total events $N$ to pass some requirements. If $N_{\mathrm{sel}}$ of the total $N$ are selected, the sample efficiency is defined as $\hat{\epsilon} \equiv N_{\mathrm{sel}}/N$. An analytic estimator of the uncertainty on the efficiency is obtainable via Binomial statistics:[1]

$$\hat{\sigma}^2(\hat{\epsilon})_{\mathrm{B}} = \frac{\hat{\epsilon}(1-\hat{\epsilon})}{N} \, , \tag{6}$$

which has the pleasing feature that the one-sigma error band $[\hat{\epsilon} - \sigma(\hat{\epsilon})_{\mathrm{B}} \ldots \hat{\epsilon} + \sigma(\hat{\epsilon})_{\mathrm{B}}]$ is always contained in the range $[0,1]$: this is semi-accidental, but conveniently logical, and sufficient for most practical purposes. (In the rare cases where it is not, a non-analytic MC-toy computation of the efficiency probability distribution is usually required.)

## 2.3 Weighted moments, mean and variance

Returning to our weighted moments, we can extend the concept of our finite set of sampled values to include the concept of *fill weights*, $w_n$. These represent rescalings of the probability of the fill with respect to the nominal $w_n = 1 \, \forall n$ implicit in the unweighted case: if $w_n > 1$ it indicates that this single fill is statistically representative of greater than one fill's worth of

---

[1]As opposed to Poisson, since the number is now fixed and we are choosing between selected or not, rather than an unknown total number with known rate of random occurrence.

probability density, and *vice versa* if $w_n < 1$ it indicates that this fill-value $x_n$ (or at least the route by which it was obtained) has been overrepresented and should be considered as worth less than one fill.

Weights can occur either for natural measurements or for simulated ones, reflecting either or both of a biased selection process or a variation to assumptions made in the measurement process. For example, on the first point, a measurement may use "prescaling" to randomly discard events/fills selected by different routes in order to limit data rates and balance contributions from the different sources, or an MC simulator may intentionally deviate its samplings from the naive probability distributions in order to again achieve a more equitable distribution of fill counts across the space of values [2]. On the second point, the values being filled may have been obtained by inference from a more fundamental set of "raw" observations: if the nominal assumptions underlying that inference are varied, the relative probability of each fill-value will change, giving a weight $w \propto P_{\text{var}}/P_{\text{nom}}$.

In short, fill-weights are a mechanism for offsetting biases in the fill-generation process in order to obtain an unbiased distribution or its moments. Hence, we must generalise our moment estimators to include the effects of fill weights (where from now on we drop the explicit summation ranges over sample indices):

$$\langle \hat{x} \rangle = \frac{\sum_n w_n x_n}{\sum_n w_n} \tag{7}$$

$$\begin{aligned} \hat{\sigma}^2(\hat{x}) &= \mathcal{B} \cdot \frac{\sum_n w_n \left( x_n - \sum_m w_m x_m \right)^2}{\sum_n w_n} \\ &= \mathcal{B} \cdot \frac{\left( \sum_n w_n x_n^2 \right) \cdot \left( \sum_n w_n \right) - \left( \sum_n w_n x_n \right)^2}{\left( \sum_n w_n \right)^2} \\ &= \frac{\left( \sum_n w_n x_n^2 \right) \cdot \left( \sum_n w_n \right) - \left( \sum_n w_n x_n \right)^2}{\left( \sum_n w_n \right)^2 - \sum_n w_n^2} \; . \end{aligned} \tag{8}$$

As for the unweighted cases, these expressions incorporate the effect of Bessel's correction, subtracting "one fill's worth" of information in the computation of the outer expectation values. This correction is captured in the multiplicative factor $\mathcal{B}$, which for unweighted statistics is $\mathcal{B}_{\text{U}} = N/(N-1)$ to effectively replace the naïve $1/N$ averaging with $1/(N-1)$. For weighted statistics what "one fill" corresponds to is encoded in the *effective fill-count*,

$$N_{\text{eff}} = \frac{\left( \sum_n w_n \right)^2}{\sum_n w_n^2} \; , \tag{9}$$

which is used by direct replacement to give the *weighted Bessel factor*,

$$\begin{aligned} \mathcal{B} &\equiv \frac{N_{\text{eff}}}{(N_{\text{eff}} - 1)} \\ &= \frac{\left( \sum_n w_n \right)^2}{\left( \sum_n w_n \right)^2 - \sum_n w_n^2} \; , \end{aligned} \tag{10}$$

and hence the final form in eq. (8). Note that the effective replacement on the denominator is now $\left( \sum_n w_n \right)^2 \rightarrow \left( \sum_n w_n \right)^2 - \sum_n w_n^2$. It is simple to verify that the effective count is invariant under uniform global rescalings of all weights, $w_n \rightarrow a w_n$, and hence if all fills have the same weight, however large or small that weight is, the effective fill-count of the sample is equal to

its naïve number of entries, $N$. In weighted statistics, $N_{\text{eff}}$ indicates the degree of statistical stability of the sample, which can only be smaller than in the unweighted (or equivalently, equally weighted) case. In extreme circumstances such as balanced fills with positive and negative weight-sign, the effective number of fills can be zero, as a useful indication that the variance estimate will be unreliable.

## 2.4  Histograms

Having established a coherent statistical picture for unbinned quantities in one dimension, we now both generalise the measured variable $x$ to a vector *variable-space* $\Omega$ composed of vectors $\omega$ and with a differential volume element $\mathrm{d}\Omega$, and partition that space into a disjoint (sub)set of *bins*, $\{\Omega_b\} \subset \Omega$.

The generalisation of the variable dimensionality has little profound effect beyond extending the possible set of weighted moments to $\langle \omega^{(i)} \rangle$ and $\langle \omega^{(i)} \omega^{(j)} \rangle$ for all dimension indices $i, j$ in $\Omega$. In general the cross-terms of this generalised second moment encode variable correlations, which can be transformed into a sample-covariance matrix via generalisation of the moment-construction for self-variance, including the weighted Bessel factor of eq. (10):

$$
\begin{aligned}
\widehat{\Sigma}_{ij} &= \langle \omega^{(i)} \omega^{(j)} \rangle - \langle \omega^{(i)} \rangle \langle \omega^{(j)} \rangle \\
&= \mathcal{B} \cdot \frac{\left( \sum_n w_n \omega_n^{(i)} \omega_n^{(j)} \right) \cdot \left( \sum_n w_n \right) - \left( \sum_n w_n \omega_n^{(i)} \right) \cdot \left( \sum_m w_m \omega_m^{(j)} \right)}{\left( \sum_n w_n \right)^2} \\
&= \frac{\left( \sum_n w_n \omega_n^{(i)} \omega_n^{(j)} \right) \cdot \left( \sum_n w_n \right) - \left( \sum_n w_n \omega_n^{(i)} \right) \cdot \left( \sum_m w_m \omega_m^{(j)} \right)}{\left( \sum_n w_n \right)^2 - \sum_n w_n^2} \, .
\end{aligned}
\tag{11}
$$

The bin partitioning introduces a new concept: where the unbinned moments converged to summary statistics of the entire probability density function $f(\Omega) \equiv \mathrm{d}P/\mathrm{d}\Omega$, the moments in each bin $b$ converge to the summary properties of that bin's variable-space partition, i.e.

$$
\langle \omega^{(i)} \rangle_b \equiv \int_{\omega \in \Omega_b} \omega^{(i)} f(\omega) \, \mathrm{d}\Omega
\tag{12}
$$

$$
\langle \omega^{(i)} \omega^{(j)} \rangle_b \equiv \int_{\omega \in \Omega_b} \omega^{(i)} \omega^{(j)} f(\omega) \, \mathrm{d}\Omega \, .
\tag{13}
$$

For consistency, all moments need to converge to their unbinned values when the partition is expanded to include the whole space, and to converge to the differential properties of the probability density function itself when the partitions are made infinitesimally small, $\Omega_b \to \mathrm{d}\Omega(\omega)$. In addition, integrating out dimensions of the variable space (i.e. combining bins along axes) must for consistency converge to the same result as having originally constructed a lower-dimensional or less finely binned partition of the space: this is guaranteed by the linearity of the sums over the sample indices in the statistical-moment definitions.

This perspective illustrates the fundamental connection between differential histogramming and differential calculus: a statistical histogram is not just a collection of fill counts, but a discrete approximation to a continuous probability or population distribution. The ideal differential distribution (or density) is $\mathrm{d}P/\mathrm{d}\Omega$ in the case of a probability density, and $\mathrm{d}N/\mathrm{d}\Omega$ in the case of a population density; histograms approximate this by converting the differentials to finite deltas, $\Delta P/\Delta \Omega$ and $\Delta N/\Delta \Omega$, though the differential analogy is often preserved by use of the infinitesimal d symbol in plot labelling.

We note in particular that the *bin measure* $d\Omega$ or $\Delta\Omega$ representing the volume element of the bin is a crucial consistency element in constructing a histogram's bin values: to preserve the density estimate, the width (or generally volume) $d\Omega$ of the containing bin must be divided out so the $\Delta \to d$ limit converges. This is particularly important as in general it is not desirable for finite bins to have the same width: for the statistical relative uncertainty on bin populations to be equally distributed across the histogram, bins *should* be made larger in regions of low density, and narrower (until the variable-resolution limit) where there is high sample density. With non-uniform bin sizes, failing to divide by the bin measure distorts the distribution away from its physical shape.

Should one wish to compute and display the actual bin population, one would need to either – ideally – use a discrete binning expressed in terms of finite probabilities rather than densities, or as a workaround multiply each density bin by its fill-volume. In the absence of a more official name for this object, and reflecting its typical use, we refer to this not as a histogram, but as a *bar chart*.

## 2.5 Profiles

Our final statistical preliminary is to slightly generalise this concept of a histogram. In the previous section we defined the binning partition across the entire variable-space $\Omega$. But a useful class of histogram mixes binned and unbinned variable subspaces, allowing characterisation of the unbinned dimensions $\Upsilon$ via their moments as projected into each partition of the *bin-space* $\Theta$.

These partially binned objects are known as *profiles*, effectively histograms in the bin-space with augmented moment content in each bin. Conceptually, they are very useful objects for scientific work as they allow statistical aggregation of finite samples into "independent variable" bins $\theta \in \Theta_b$, while characterising the mean dependence of the unbinned dependent variables $\mathbf{y}$ on $\theta$. Again, the limiting behaviour must be that for infinitesimally small bins and infinite sample statistics, the fully differential relationship $\mathbf{y}(\theta)$ is obtained. Aggregation of bins and reduction of the unbinned space to lower-dimensions must again, as with aggregation of the binned subspace, give the same result as having originally constructed the lower-dimensional profile – again guaranteed by the linearity of the statistical moments.

In general a profile's unbinned space, $\Upsilon$, can be multidimensional, but this introduces an ambiguity as to the resulting canonical bin value – each profile bin effectively contains the histogram-type integrals of fill weights as well as the set of moments corresponding to a multivariate Gaussian[2] distribution between the unbinned variables. For definiteness, we restrict ourselves to a single-dimensional unbinned space $y$, whose relevant moments are $\langle y \rangle$ and $\langle y^2 \rangle$. Conventionally the profile canonical bin value is the mean $\langle y(\Theta) \rangle$ as a function of the binned coordinates, and rather than the standard deviation of the unbinned distribution, the *standard error* $\hat{\sigma}_{\bar{y}}(\theta) = \hat{\sigma}_b/\sqrt{N_b}$ is used as the nominal uncertainty, where $N_b$ is the effective sample count in bin $b \supset \theta$.

With this, we have finished setting the scene: while not complex in grand terms, the importance of consistency constraints and the potential for high-dimensional and combinatoric complexity is clear. In the next section we consider the implications for a computational implementation that preserves these principles, and establish core design goals for an implementation.

---

[2] Or more generally an elliptical distribution.

# 3 Design principles

YODA is a package for creation and analysis of statistical data, particularly various flavours of histogram, written primarily in C++ and programmatically usable from C++ and Python. The development and developers of YODA emerged from the sub-field of Monte Carlo event generator analysis and tuning [3] in high-energy physics (HEP), which places certain requirements and emphases on its functionalities, but the library is deliberately agnostic of any particular application.

A brief list of these requirements is useful to frame our design choices, though we will see that some of these ideas naturally led to generalisations beyond what we present here:

**Differential consistency:** a histogram is fundamentally not just a list of (weighted) fill-counts, but a binned best-estimate of a continuous distribution. Optimal estimation requires non-uniform binnings in proportion to expected probability density, and so it is crucial that these different fill-space volumes be divided out for each bin when estimating the binned density, i.e. taking the $f(\mathbf{x}) \equiv \mathrm{d}P(x)/\mathrm{d}\mathbf{x}$ notation literally.

**Continuous aggregation:** for studying extremely large datasets as is often the case in HEP, it is not feasible to perform histogramming in one pass over a full set of fill values stored in memory. Over thousands of fill observables and potentially billions of events, this mode (as exemplified by e.g. `numpy` and `Matlab`) is not computationally viable. Instead, many HEP and similar large-data applications require a mode in which histograms are "live" summary objects to which new entries/fills can be continuously added.

**Weighted statistical moments:** the key summary statistics needed from histogram bins are the bin value (as defined by the type of histogram being used) and its statistical uncertainty, but also the mean and (co)variance of the fill-distribution within the bin's coordinate space. The fills are in general weighted according either to how they were sampled or manipulated, and so histograms need to store the weighted statistical moments required to compute the key summary statistics of their bins.

This picture also provides a consistent way to view the extended "profile" histogram type: in addition to storing the statistical moments of the fill weights and the fill dimensions, a profile bin also stores the moments of further unbinned dependent values, $\mathbf{y}$.

**Integral consistency:** Statistical moments, as defined from the pdf function in eq. (1), are intrinsically integral quantities, computed via marginalisation across the variable-space $\Omega$, or subspaces of it. This integral property maps into the sampled moments and summary estimators. As such it should be possible to reconstruct their full integral values (or the best available estimate from finite fill statistics across the whole space) by composing together their estimates in subspaces, e.g. in different bins. A consistent computational statistics library using binned quantities should be able to project high-dimensional binnings into lower-dimensional ones (for example, constructing a binned profile along one axis of a higher-dimensional histogram) without biasing integral quantities, e.g. by replacing them with discretised bin-centre estimates in the high-dimensional binning.

**Separation of style from substance:** a histogram is first and foremost a data object representing the statistical properties of the fills recorded into its bins, rather than any particular rendered representation of that data. One should be able to ensure the invariance of statistical data while varying the plotting style.

**Separation of binning from bin-content:** this latter point can be more generally viewed as motivating a separation between *live* and *inert* classes of data-object: the former are the

statistical objects tracking the evolution of summary moments (and permitting further data-taking) while the latter are a specific representation of chosen data facets into a set of "values and uncertainties" for unambiguous plotting or summarising. But once this thought-process is underway, a further design clarification arises: the mechanism for aggregating multi-dimensional coordinate ranges into discrete bins is useful independently of the statistical moments they contain, and a suitable generalisation makes it possible to unambiguously implement live and inert binned data types using the same abstract binning framework.

**User friendliness:** while any sort of programming would be regarded as user-unfriendly in many circumstances, programmatic data-analysis is the norm in quantitative science, and the programming interfaces of libraries cover a wide spectrum often between powerful-but-intimidating and easy-but-inflexible. Our motivation is to provide a "clean" programmatic interface expressed in terms of statistical and data-analytic concepts and hence well-matched to the goals and skill-sets of data scientists, rather than emphasising language technicalities or requiring systems-programmer levels of programming familiarity.

The continuous-aggregation design principle excludes many existing data-analysis libraries from contention for HEP and other large-data tasks. The most prominent packages for continuous-aggregation histogramming are the HEP ROOT [4] framework and arguably the Boost.Histogram [5] library.

The former is heavily used in HEP for histogramming in low dimensionality with limited axis and bin-content types, but conflates live and inert types (allowing for inconsistency with fill history), and data and presentation uses.[3] ROOT is also a very large monolithic framework with many dependencies, including many features beyond statistical analysis and expecting to be the controlling element in applications rather than called as a utility library: these can be obstacles to use in many applications.

By contrast, Boost.Histogram focuses on an abstract implementation of binning in a style reminiscent of the C++ standard library, and making explicit use of advanced language features and abstractions distanced from statistical terminology. It is hence a powerful tool, but the interface imposes a barrier for less technically adept C++ users. In the design of YODA 2 we attempt to negotiate a third path in which advanced language features (similar to the Boost.Histogram approach) are *internally* used to enable high levels of abstraction and to enforce statistical consistency and type-safety, but also providing a C++ (as well as Python) user-interface in which this complexity is hidden and the application alignment emphasised. YODA is also intentionally limited to statistical analysis only, requires no library dependencies for core C++ operation, and operates purely as a class library rather than a stateful controlling framework – choices explicitly made to assist embedding into applications.

## 4 Experience from YODA version 1

The top-level design goals stated above were partially established at the time of the YODA version 1.0 release in 2013, e.g. with fully correct (to second order) statistical-moment combination a core principle of the design. But deployment experience in the meantime, and in particular extension to multidimensional histograms binnings, revealed structural issues which motivated the rewrite described in the following section:

**Wrapper-type issues:** A *good* design decision was to isolate statistical moment calculations into a set of `Dbn` (distribution) classes for each dimensionality. For example, a weighted

---

[3]These should be read not as a criticism, but as a pragmatic design choice on which YODA takes a different path.

"counter" type was just a zero-dimensional distribution which tracked sums of weights and squared weights, each histogram bin was implemented around `Dbn1D`, one-dimensional profile histograms contained a `Dbn2D` in each bin, and so-on. But each of these bin-wrapper types required repeating and mapping the majority of `Dbn` interface methods, which with the combination of binning and `Dbn` dimensionalities created a significant maintenance overhead. The need for actual wrapper types to be stored also bloated the analysis-object memory requirements and fragmented the memory layout relative to a contiguous array of the fundamental `Dbn`s.

**General irregular binnings:** YODA 1 had intrinsic, first-class support for non-uniform binnings, as opposed to an awkward add-on, as in many HEP applications the probability density over an observable's range of interest varies by orders of magnitude: with non-uniform binning as the only reasonable strategy to balance resolution with statistical precision, it needed to be easily invoked. But the general-dimensionality extension of non-uniform binnings is irregular tilings of the binning space, and here the ambition proved a step too far. Even in two dimensions, computing the (non-)overlap or commonality of two binnings was a significant computational challenge, and more complex in higher dimensionalities, for little or no practical gain.

**Overflows:** Noting the unphysicality of bin-property requests to underflows and overflows in the ROOT histogram classes, YODA 1 implemented overflow bins as bare `Dbn` types, avoiding interpretation as standard bins with widths/areas, etc. In one-dimensional data types this was no issue, as there were only two overflows to be dealt with, i.e. the underflow and overflow below and above the binned range respectively. But in 2D (and higher dimensionalities), distinct overflow distributions were needed above and below every row and column of the in-range binning: an exponentially more complex problem, and even an uncomputable one given the promise of general irregular binnings and gaps. YODA 1 never provided fully functional overflows in multiple dimensions, blocking for example the dimensional reduction of a 2D histogram into a projected 1D histogram or profile.

**Bin gaps:** Motivated by gaps in binning[4] in legacy data records, YODA 1's binnings were implemented as a list of explicit bin objects which knew their own edge locations, information then duplicated by the axis objects responsible for locating the bins given a set of fill coordinates. A hidden "total distribution" object was required to track fills that landed in the gaps but should still contribute to overall normalization. This structure also lended itself to ability to sequentially add and remove single bins, which then required a locking mechanism for consistency to ensure that binnings could not be altered once filling had begun. Again, this was significant unnecessary complexity and maintenance difficulty.

**Mismatched live/inert types:** YODA 1 did clearly separate live histogram types from inert types, largely informed by necessity due to the limited information available in reference-data resources such as the HEPDATA publication-data database. But all inert types were implemented as "scatter" objects: a set of $n$-dimensional points and error bars, without any concept of binning. This made comparisons between inert scatter types and live types awkward, with the binning needing to be heuristically reconstructed from the absolute positions of the ends of the error bars, for example if using a reference-data object to set the appropriate set of bin edges on a new live object. This process worked in most circumstances, but was not efficient, and was based on guesswork and convention rather than a type-safe approach. In addition, as an inert format is naturally the type to

---

[4]Not just empty bins, but unreported regions in the fill range.

be used for data preservation and interpretation, more structured statistical data such as uncertainty breakdowns (for various classes of statistical or systematic error) was retrofitted incoherently on to the scatter types. This involved more arbitrary or conventional heuristics, rather than a fully coherent design, and opportunities for inconsistency in conversion between pointwise and whole-object views of the encoded correlations.

## 5  Implementation

All these issues, and myriad smaller ones, led to a wholesale rethink of the YODA implementation in order to better meet the now clear requirements of a general and fully consistent histogramming library. At the same time, it remained paramount that the resulting API be usable by a typical physics programmer, via compact and expressive data-handling code either in straightforward C++ or via the Python wrapper package.

As the mathematical principles of consistent multidimensional statistics, and the resulting design goals of the redesign, are expressible declaratively at compile time, there is a natural synergy between this application and the template-metaprogramming methods of modern C++. As well as guaranteeing conceptual consistency and type correctness, and avoiding the code-duplication maintenance woes of the original YODA release series, a template-oriented approach is consistent with engineering for high-performance applications, rather than deferring dimensionality calculations to runtime. Accordingly, the YODA 2 architecture is heavily based on modern C++ 17 template methods [6], and will evolve in-line with the language's focus on making such methods more powerful and accessible [7].

### 5.1  Bin partitioning

YODA 2 builds its partitionings of the fill space by composition of multiple one-dimensional binnings. These 1D binnings are implemented via a unified `Axis` class (a refinement of YODA 1's `Axis1D`) that is templated on the edge type. The default implementation assumes a discrete binning with one bin for each edge label and an additional catch all-else "otherflow" bin at index `0`, while the more traditional continuous-value axes are provided as a template specialisation triggered when the template type satisfies the `std::is_floating_point` trait.

The floating-point specialisation implements $N-1$ bins for $N$ edges, as each bin is enclosed by a lower and upper bin edge with a shared edge between two neighbouring bins. It also adds an underflow bin between `-inf` and the lowest finite edge as well as an overflow bin between the highest finite edge and `+inf`. This infinite-range binning ensures consistent slicing behaviour in higher dimensions across multiple axes and their respective sets of under- and overflow bins. YODA actively uses the IEEE 754 floating-point standard's [8] `inf` and `nan` values, and their standard combination rules with normal floating-point values to express the range of computable binning and bin-value quantities rather than necessarily treating these overflow values as errors.

Helper methods are provided for 1D binning-specification, including Matlab/numpy -influenced `linspace` and `logspace` functions, and a general `pdfspace` for distribution of $N$ bins proportional to an arbitrary probability density.

The global fill-space $\Omega$ is partitioned into bins via the outer product of independent 1D axes, giving a rectilinear (but generally non-uniform) binning grid as illustrated in Figure 1. A single bin on one independent axis corresponds to a family of bins in the remaining dimensions of the fill-space: These mappings can be efficiently computed, permitting general *slicing* and *marginalising* across global fill-space.

This functionality is implemented in a dedicated `Binning` class, which can also translate between the tuple of local bin indices (one for each axis) and a global bin index (in $\Omega$-space)

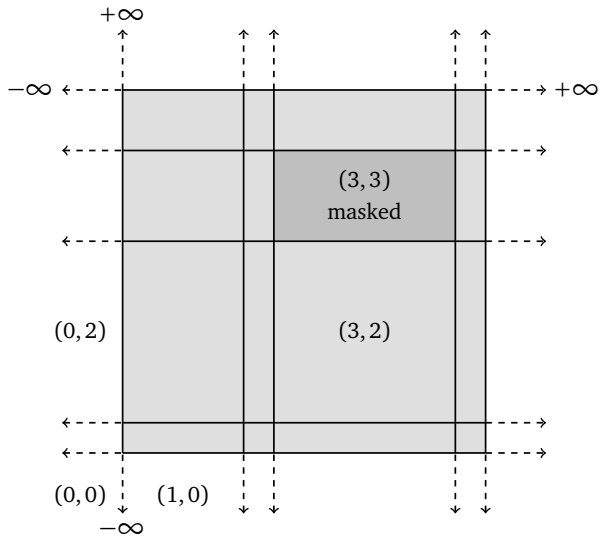

Figure 1: Illustration of the YODA 2 generalised infinity-binning scheme for continuous axes. The grey-shaded region represents in-range bins, with the white regions extending to positive and negative infinity being the overflow bins. Local-index tuples are annotated on several bins, illustrating the zero-indexing of axis-local indices when including underflow and overflow bins. Bin masking is shown at index $(3,3)$.

via `globalToLocalIndices` and `localToGlobalIndex` methods, using a standard nested-stride mapping scheme.

The treatment of underflows and overflows as fully fledged bins means that the local indices start at 1 for the in-range bins. Gaps in fill-space are supported via *bin masking* rather than bin erasure, ensuring that the underlying binning always counts all fills and that total fill statistics can be computed by integration over bins, rather than losing information in bin-gaps (or needing to maintain a separate "total" counter as in YODA 1).

This simple factorisation of per-axis binning structures avoids the issues with high-dimensional bin-overlap computation and necessity of fill-locking experienced with YODA 1's naïvely ambitious arbitrary-binning model. Should a user wish to use irregular bin tilings across the global fill-space, these are not directly supported, but can be achieved via post-processing given a suitably granular initial binning.

The factorised-binning model is also efficient in regular use, as the global bin-index is computed in fixed time from the set of local bin-indices along each axis; these in turn are computed in an optimised fashion using a `BinEstimator` object created along with the binning. This pre-assesses whether a linear search strategy, or a binary-search in either linear or log-space will be the most efficient for bin discovery given a fill coordinate, such that the asymptotic complexity of $D$-dimensional bin-lookup over $N_{bin}$ bins per axis is $\sim D \ln N_{bin}$. For use-cases where the same coordinates will be used repeatedly on equivalent histograms, the bin index can be cached to avoid duplicated bin-search computations. The global index can then be computed and used in $\mathcal{O}(1)$ time for access to contiguous storage corresponding to the bin contents.

## 5.2 Bin content

The *content* of the partitioned bins is generalised in YODA 2 to allow storage of arbitrary types within binnings, paving the way for discrimination between simple storage of inert quantities (e.g. a float-valued binned efficiency-map) and more tightly integrated live types as discussed

in Sections 3 and 4.

To maintain the impression of bins as distinct objects looked-up within the binning, and with awareness of their location as well as content, a templated `Bin` class is provided to wrap around the bin content (by inheriting from and augmenting it with bin-location information such as indices, edges and geometric midpoints) and to provide a link to the parent `Binning` object.

Any given bin covers a volume element `dVol()` in $\Omega$ fill space, given by the product of all continuous-axis bin widths, which can become infinite if a bin falls into the under- or overflow range of a continuous axis. Semantic `dLen()` and `dArea()` aliases are provided in 1D and 2D, respectively. In mixed continuous–discrete fill space, the discrete bins default to a unit width to allow for the fill-space element to be finite. The concept of "height" that was present in YODA 1, motivated by typical graphical histogram representations, has been removed from v2 as there is neither a guarantee that the stored content is arithmetic, nor that the traditional plot representation is relevant.

Access to axis-specific quantities is via templated accessor methods, e.g. a bin `b`'s minimum-value edge on the $d$'th axis is accessed like `b.min<`$d-1$`>()`, noting the zero-indexing of the axis indices. The curiously recurring template pattern (CRTP) is used to mix in axis-specific method names for the first three dimensions (`xMin()`, `yMax()`, etc.) and to reduce the amount of code duplication.

### 5.2.1 Live content

The `Dbn` distribution class from YODA 1 has been generalised to arbitrary dimensions by templating it on the dimensionality. It tracks the number of events, the sums of weights and squared weights, and the weighted first- and second-order moments of axis position (including mixed moments e.g. $\sum_n w_n x_n y_n$) which allow computation of the means and variances along each independent axis as well as the general covariance matrix. The vector mean position of weighted fills within a bin is referred to as the *bin focus* and is exposed through the bin interface, meaning that there is no need to approximate a bin's effective statistical location as being at its geometric midpoint.

The `Dbn` class has a `fill` method that accepts the next weight, a coordinate for each dimension, and an optional *fill fraction*. The concept of a fractional fill is motivated by numerical instabilities in quantum chromodynamics, where the "perfect resolution" of a sharp bin-edge can introduce non-cancellation of infinities (or in computational reality, non-cancellation of large, oppositely-signed weights in correlated fills); by spreading a fill over neighbouring bins, numerical stability can be restored. While motivated by quantum-mechanical issues, the additional concept of a fill-fraction in addition to the more established fill-weights may be of use in other domains.

### 5.2.2 Inert content

Representation of binned inert content is a new feature in YODA 2, reflecting the conceptual mismatch in YODA 1 between binned live types and point-based representation of inert reference-data.

A new `Estimate` class has been introduced, consisting of a single central value as well as an optional dictionary of labelled error pairs, representing identified sources of nominally 1-sigma uncertainty (from which a systematic covariance can be constructed between bins). The signed error pairs are understood to be the results of respectively downward or upward shifts in a correspondingly named nuisance parameter; the empty string is interpreted as a user-supplied total uncertainty pair. Error labels starting with `stat` or `uncor` are treated as uncorrelated in arithmetic operations, while all other error labels result in a fully correlated treatment of the

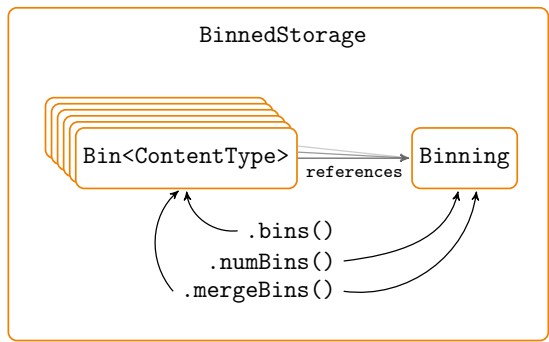

Figure 2: Illustration of the internal `BinnedStorage` structure in YODA 2. Each bin is a light wrapping of the raw content type, with bin-location information made accessible via mixed-in methods making use of the linked `Binning` object. Methods implemented on `BinnedStorage` may rely on either the binning data, the content data, or both.

corresponding error pairs. If a different behaviour is desired, an alternative `regex` parameter can be provided to processing functions such as `divide`, to trigger the uncorrelated treatment. Various convenience methods are provided to translate the default down/up representation into generally asymmetric negative/positive error bars induced on the estimate value. The total uncertainty is by default given by the sum-in-quadrature of all negative and positive components, unless a user-provided total uncertainty pair is present.

## 5.3 Combined partitioning and content

The types introduced so far permit histogramming to be implemented using separate bin-location and bin-content facilities, but the user would need to manually maintain consistency between them. To ensure this automatically, a generic `BinnedStorage` class is introduced, illustrated in Figure 2. This type contains both a `Binning`, constructed from a C++ parameter pack of the edge types defining the fill space, and stored bin-content as a contiguous array of `Bin`-wrapped content types. The `BinnedStorage` manages its fill space, providing access to individual bins via a `bin` method which accepts the global bin index or the array of local indices, a `binAt` method for locating the bin via a set of $\omega$ coordinate values, and a `bins` method which returns a wrapping vector type that can elide overflow and masked bins depending on optional arguments. The `BinnedStorage` also ensures consistency when redefining binnings, e.g. via the `mergeBins` method which simultaneously eliminates edges across a range of bins, and merges the corresponding content objects.

This structure, however, is still too generic for user-facing standard histograms. In order to achieve a live, update-able specialisation, an additional `FillableStorage` inheritance layer is introduced, providing the means to update its internal statistics using the C++ adaptor pattern. A *fill adaptor* type can be passed in by the user, allowing customisation of the *fill* concept depending on the specifics of the bin-content type in question. Whenever the user calls `fill` on the object, passing a set of coordinates and an optional fill-weight and fill-fraction, the `FillableStorage` uses its contained `Binning` object to identify the targeted bin, then passes its global index to the fill-adaptor object, which handles the specialised updating of the bin's content. At the end of the `fill` call, the global bin index is returned. If any of the coordinates is `nan`, it is in general not possible to identify the corresponding bin and so a `-1` is returned, and the `nan`-fill recorded. If a bin can be identified but is masked, the bin statistics will still be updated and the position returned. This gives the user more flexibility to change

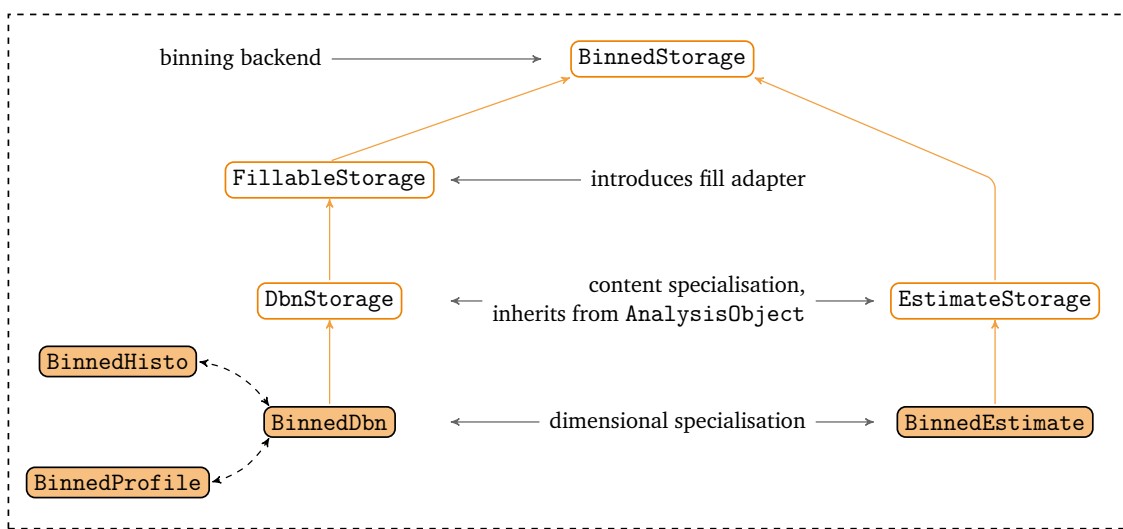

Figure 3: Internal binned-type class inheritance structure in YODA 2. The diagram illustrates how several layers of inheritance and template-specialisation (orange arrows) are used to build parallel live and inert data-types, with higher layers more generic and lower ones more specialised and user-facing for standard statistical operations, in particular the specialised aliases connected via dashed lines on the bottom-left.

their mind about unmasking or merging bins at a later time. The `FillableStorage` also adds the fill-dimensionality as an additional template parameter, allowing in general the `fill` method to accept more coordinates than the binning dimensionality, e.g. enabling a unified implementation of histogram and generalised-profile live binned data-types.

## 5.4  Standard histograms and profiles

Further stages of specialisation, illustrated in Figure 3, are used to reduce this highly general picture of binned and fillable data-storage to the familiar statistical types discussed in Section 2, presenting a user-facing view of the most commonly used statistical machinery. The desired live and inert types are respectively implemented as a `FillableStorage` containing `Dbn` objects and a `BinnedStorage` containing `Estimate` objects. Again inheritance and template special-isations are used to provide features and reduce code duplication, with derived `DbnStorage` and `EstimateStorage` types adding whole-object facilities such as integrals over the live `Dbns`, or area-under-curve or estimate-averaging facilities on the collections of inert binned estimates. At this point, with the content types fully specified, the data-objects also acquire an inheritance relationship from the virtual `AnalysisObject` base class which provides a metadata storage utility and integration with the I/O persistency systems to be described in Section 6.

The very final step is to provide some user-friendliness refinements for the most familiar types: the `BinnedEstimate` and `BinnedDbn` inheritance layers employ the CRTP to mix in axis-specific method names for the first three binning dimensions, e.g. `xEdges` or `sumWY2`.

Various aliases are defined to enable convenient shorthands, with the whole user-facing type family shown in Figure 4:

- `BinnedHisto` takes a parameter pack of edge types and is an alias for a `BinnedDbn` with its fill dimensionality given by the number of elements in the parameter pack.

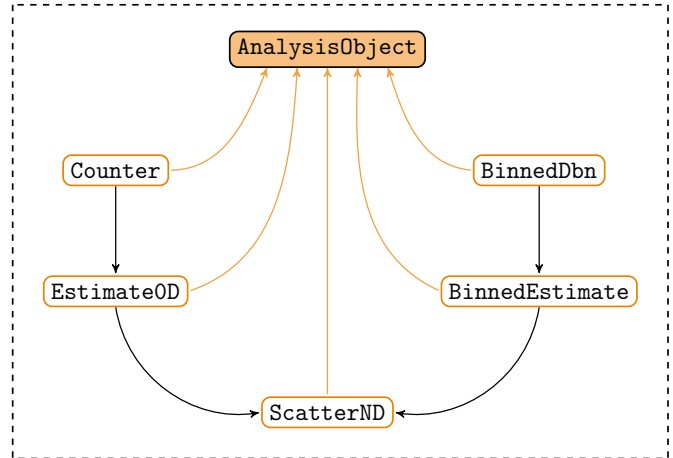

Figure 4: User-facing class inheritance structure in YODA 2. The orange lines show that all these types inherit from the persistency and metadata features of `AnalysisObject`, and the black arrows indicate the direction of live-to-inert type-reductions. Direct type reductions to `ScatterND` from `Counter` and `BinnedDbn` are also allowed, bypassing the intermediate `Estimates`.

- `BinnedProfile` takes a parameter pack of edge types and is an alias for a `BinnedDbn` with its fill dimensionality given by the number of elements in the parameter pack plus one, for the unbinned axis.

- `HistoND` and `ProfileND` are both templated on a positive integer that is internally translated into a corresponding number of `double`s, as a shorthand for a histogram or profile with only continuous axes, respectively.

- For the first three dimensions, more legible aliases `Histo1D`, `Profile2D` etc. are also defined, largely for familiarity with the previous YODA 1 implementations.

The full suite of natively supported user-facing types inheriting from `AnalysisObject` is illustrated in Figure 4. The `BinnedDbn` and `BinnedEstimate` are accompanied by unbinned `Counter` and `Estimate0D` objects which have no fill-space and hence correspond to the unbinned summary statistics from the start of Section 2; these are implemented separately due to their simplicity.

Timing performance tests were made using a similar methodology to Boost.Histogram (BH), using 1M samplings per dimension from a unit Gaussian located at $\mu = 1$. These were filled into histograms of dimensionality 1 to 5, each axis containing 20 bins either uniform in $[-2, 2]$ or logarithmic from $[0.02, 2]$, ensuring the non-linear search and overflow features were tested. Linear scaling of fill time was observed in single-threaded running on a 4.0 GHz CPU, from 0.19 (0.21) secs in 1D to 0.94 (0.99) secs in 5D for 1M fills into linear (logarithmic) binnings respectively. While histogramming is rarely a performance bottleneck and this speed is sufficient for analysis use-cases, it is slower than BH to the extent comparisons can be made given the difference between full-moments histogramming in YODA 2 vs integer-count fills in the BH benchmarking. This remains an area for possible performance optimisation.

## 5.5 Type and dimensionality reductions

At the end of a run, the live types can be reduced to their inert equivalents and can be subjected to further post-processing, e.g. in order to combine systematic variation runs. Inert types are also

the natural output from object-combining operations such as histogram division or efficiency computation (differing by uncertainty treatment). The inert types lend themselves naturally to representing the experimental cross-section measurements available on HEPDATA [9], and the built-in `covarianceMatrix` method of the `BinnedEstimate` class can be used to translate these error breakdowns into a covariance. When reducing a live type to an inert type, masked bins are skipped, thereby retaining the default `nan` of the corresponding `Estimate` objects contained within the `BinnedEstimate`, which can then be interpreted as a visual gap by plotting scripts.

Both live and inert types can also be reduced to the `ScatterND` class which contains an array of `PointND` objects – a set of coordinates and negative/positive uncertainty pairs in $N$ dimensions, which lends itself directly to the concept of a marker on the canvas. When converting an `Estimate` to a `PointND`, the total uncertainty is transferred and the error breakdown is lost. Bin-focus information is lost from live objects if converting to a scatter via an estimate, as the inert binning-content has no such concept, but can be preserved if reducing directly from a live type to a point-based scatter.

All inert types, including scatters, can also be handed functors such as C++ lambda functions in order to apply arbitrary transformations on the inert statistical distribution. These will operate on the underlying `PointND` or `Estimate` objects, with consistent non-linear propagation to both values and errors, with appropriate distinct treatment of correlated and uncorrelated error-source in the latter case.

It is also possible to switch between different types of `BinnedDbn`, typically by reducing either the fill- or binning dimensionality in the process. For instance, a histogram with $N_{ax}$ binned axes can be created from a profile with the same axes by simply dropping the unbinned axis using the `mkHisto` method. Marginalisations across a given binned axis can be achieved using the `mkMarginalHisto` and `mkMarginalProfile` methods, which also reduce the dimensionality of the contained `Dbn` type, along with the binned dimensionality. In addition, it is possible to *slice* a `BinnedStorage` along one of the binned axes into a vector of `BinnedStorage` objects that are have their binning dimensionality reduced by one unit. The resulting vector will have one element for each bin along the axis that is being sliced over by calling `mkHistos`, `mkProfiles` or `mkEstimates`.

Finally, note that because the underlying `BinnedStorage` can contain arbitrary types, it is even possible to construct *histogram groups*, implemented as a `BinnedStorage` containing more `BinnedStorage` objects. This can be useful when trying to represent a 2D histogram as a group of 1D histograms connected via an implicit group axis.

## 5.6   Python interface layer

A Python interface is provided through Cython bindings to the underlying C++ implementation. The Cython source files are automatically generated from a script at build time up to a given number of dimensions and a set of axis edge types. Note that full support of arbitrary dimensions like in C++ due to its compile-time polymorphism is currently not possible in Python due to its dynamically-typed nature. Whenever lists are returned by a method, an automatic conversion to numpy arrays is attempted first with a fall-back to the default Python list type in case numpy cannot be found. "Pythonic" customisation of the API has been avoided, as the experience of YODA 1 was that distinct APIs for each language was more confusing than helpful.

## 6   Data exchange and persistency

The `AnalysisObject` type is the key link between in-memory YODA data-objects and being able to read and write persistent representations of those objects. There is no requirement to

do so: it is perfectly valid to work with in-memory YODA objects only, and unlike e.g. ROOT there is no mandatory binding of histograms and other data objects to a global I/O registry, but of course in practice much of the time users do need to be able to write out their data one way or another.

## 6.1 Attributes and I/O

The `AnalysisObject` base class provides a generic "attributes" system for storing arbitrary metadata. The fundamental attribute storage is (for now) string-based and hence not designed for high-performance storage of additional numeric data, but does provide a very useful way to store information such as normalization-scaling history, original data types (in the case of live-to-inert type reductions), plotting directives such as axis labels or legend titles, and a unique Unix-like absolute path for each object. The Python interface uses the PyYAML package [10], or the `ast` standard-library module to automatically convert the types of e.g. numerical lists stored as attributes.

The object paths are used by format-specific `Writer` classes to identify the persisted objects, and as the keys of a map returned by a singleton `Reader` class. A type register is used to help bridge the gap between the C++ compile-time templates and the desire to dynamically declare object types. The most frequent object types and set of axis edge types are preregistered when the `Reader` singleton is first instantiated and can then be augmented at run time with more complex types using the `registerType<type>()` method.

I/O operations can either be performed by explicitly instantiating these I/O manager objects, or implicitly via unbound `write` and `read` functions. At present, a structured-text format with support for gzip I/O is the main YODA persistency scheme, extended for version 2 to store bin-edges in a non-duplicating way for improved efficiency; other supported formats include a more readily human-readable representation for inspection and a "flat" format for uniform inspection of data-objects as inert scatter types. Only the structured-text format is currently supported by the `Reader` interface, and support for the defunct AIDA XML data format has been dropped; an HDF5 read/write format is planned [11].

Inspection and manipulation of YODA data files is aided by a suite of command-line tools including `yodals` for listing (in several levels of detail) file contents, and converter scripts e.g. `yoda2root` and `root2yoda`, and a `yoda2yoda` utility mainly useful for allowing users to filter (both positively and negatively) which objects are copied to the destination file.

In this supporting tool-suite and associated tools, we have used natural extensions of the Unix-path concept to index bins or points within the objects as well via suffix operators: `/my/histo` refers to a histogram, `/my/histo#3` to its bin with global-index `3`, `/my/histo2#3,4` to a bin in a 2D histogram via local indices, `/my/histo2@1.5,10.4` for coordinate-based indexing cf. the `binAt` method, and in general ranges of bins via e.g. `/my/histo#3@25`. A convention also in use is of square-bracketed path suffixes (before any bin specification) to indicate systematic variations on a nominal data object, e.g. `/my/histo[SCALE_VARIATION]`. We argue that this unified approach to histogram and bin identifiers is a generally useful concept for binned-data analysis tools and welcome wider uptake or discussion.

## 6.2 Serialisation and MPI syncing

The `AnalysisObject` additionally provides virtual methods names `serializeContent` and `deserializeContent`, to allow for translation of its derived object types to and from an `std::vector<double>`. This format lends itself better to data-gathering operations as part of MPI collective communications across different process ranks. Equivalent `serializeMeta` and `deserializeMeta` methods exist to achieve a similar (de-)construction of the object's attribute metadata to and from an `std::vector<std::string>`, which in case of estimates would

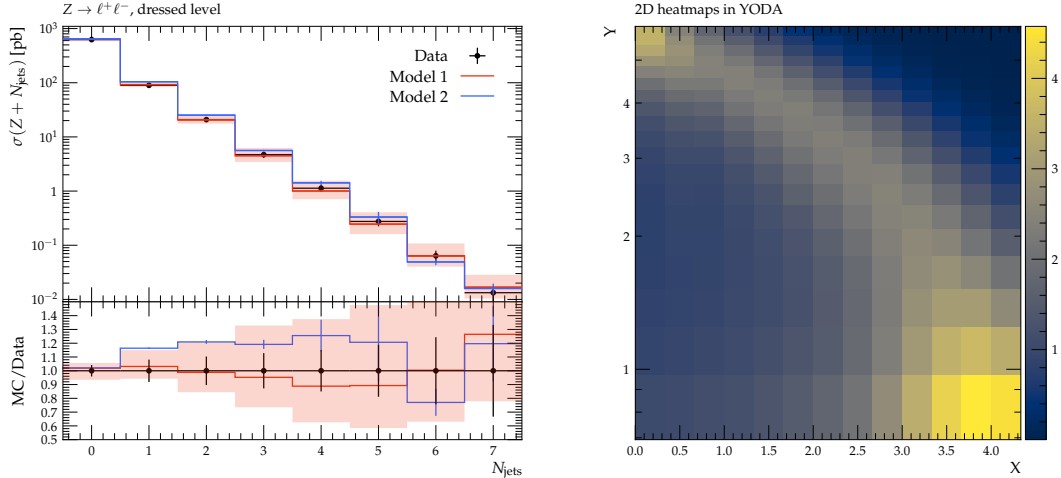

Figure 5: Example 1D and 2D plot outputs from the YODA 2 plotting interface, illustrating a mix of uniform and irregular bin sizes, and automatic use of ratio plotting for 1D data/prediction comparisons.

include the error-source labels. Serialization is primarily aimed at high-performance numerical use-cases, with a normal expectation that attributes will match between the objects being synchronised/gathered across ranks, hence the relative inefficiency of string-based metadata serialization need not be encountered in most applications.

## 7 Plotting and visualisation

YODA offers full functionality for plotting and visualising its different datatypes. The release of YODA 2 comes with a Python API plotting back-end, providing an interface with the dominant modern data-visualisation toolkit (Matplotlib ) and computational libraries (e.g. numpy ). The plotting API is designed to be quick and to produce highly customisable plots, ensuring these features can easily be interfaced to external software. For users wishing to write their own plotting code, the Python interface provides single-call access to the data typically needed by Matplotlib functions.

The YODA plotting module supports various customisations of plot features, such as the generation of ratio plots, switching between linear or logarithmic axes, the customisation of plot labels and more. This interface is designed to facilitate the interaction of YODA objects and Matplotlib , and to provide the user with out-of-the-box comparison plots in a pre-wrapped common style. As detailed below, further customisations from this starting point are possible through the Matplotlib interface.

The simplest way to plot YODA objects is with the `yodaplot` executable, which takes any number of YODA files as input. For 1D histograms, it will identify the matching object paths from the different YODA files and overlay them. This ultimately generates a single plot for each object path, i.e. the `Scatter2D` objects, where entries from multiple YODA files are shown in a comparison plot. By default an additional ratio panel is drawn, illustrating the agreement relative to the YODA file that was specified first on the command-line. The ratio panel can be disabled via a command-line flag. Two-dimensional histograms – with two independent variables and one dependent variable – are drawn on to separate canvases, so that 2D heatmaps are created for each object path (corresponding to `Scatter3D` objects) for every input YODA file. Example one- and two-dimensional plots are shown in Figure 5.

A distinctive plotting feature in YODA 2 is that in addition to the graphical outputs, for example in PDF or PNG format, the API generates a set of executable Python scripts via a "script generator" system. This is configured via a plot-specification dictionary that connects YODA data-objects to graphical representation preferences; e.g., for an analysis object `ao` read from file `fname`, a standalone Matplotlib script can be generated using

```
plotspec = { "histograms" : { fname : { "nominal" : ao }
    } }
script_generator.process(plotspec, "histo_name")
```

where `script_generator` is an instance of the script-generation manager class.

These generated scripts contain low-level plotting-calls to the Matplotlib API and can be executed to regenerate the graphical output – indeed, the graphical output of the commands is accomplished through execution of these scripts, with some optimisations using parallel processing and a shared single import of the Matplotlib library for efficiency. For maintainability, the plotting data is isolated in a separate `.py` file from the presentation commands, again providing a degree of separation between style and data.

The `yodaplot` script itself is a pedagogical example of the utilities offered by the Python API. After loading the relevant module at the start of the script, a nested loop over the input YODA files and object paths generates the Python plotting scripts through the API. In the last step, `yodaplot` executes the plotting scripts (with support for parallel processing) to generate the graphical outputs. Other software tools wrapping around the YODA plotting API include RIVET and CONTUR. In these examples, a pre-processing step collates information on the plotting style and data in a Python dictionary, which is then processed by the YODA `script_generator` function to generate executable Python scripts for plotting.

Not only does the text-based intermediate format offer a platform-independent and future-proof way to regenerate and to finesse graphical outputs, e.g. as required by iterative review processes, but the scripts are also designed to be transparent and easy to understand for the user. This design was informed by positive experience with a text-based plot-summary data format in the Rivet package's legacy TeX-based plotting system, which has now been replaced by this new implementation. As the underlying Matplotlib objects – figures, axes, lines, etc. – are accessible in the generated scripts, arbitrary customisations and refinements are possible.

## 8 Conclusions and outlook

We have introduced the YODA 2 library for consistent, high-dimensional, and flexible histogramming, expressed in type-safe modern C++. The expression of dimensionality and data-type relationships enables aggressive compile-time performance optimisation, while significant effort has been expended to ensure that the majority of technical complexity is hidden in normal usage.

A strong emphasis is placed on separation of the binning concept from aggregation of statistical moments, making possible both storage of arbitrary object types in efficient partitions of independent variables, and exact conversion between different projections of second-order statistical cumulants. Live and inert data-types are strictly separated, with one-directional conversion mechanisms and a set of simple `Scatter` data-types used as the basis for a modern Python-based plotting interface.

Persistency is currently supported to a text-based (and gzipped) custom format, with work underway to provide a more performant HDF5 format. This latter will complement current features for over-the-wire serialisation in MPI messages, making YODA 2 ideally suited for large-scale operation on high-performance computing clusters.

The YODA 2 library is already in production use with the Rivet MC analysis tool for particle physics, but is of completely general design and suited to any application in need of performant and low-dependency binned statistics or data-storage. Its programmatic usage from C++ and Python is complemented also by a set of command-line tools for dataset inspection, manipulation and combination.

Several future developments are planned, in addition to the already mentioned HDF5 data-format. A common method for encoding of systematic uncertainties is to propagate not just a single weight $w_n$ for each fill, but a family of them, $\{w_n^{(i)}\}$, where the $i$ indexes a set of *weight streams* that coherently reflect the change of the $n$'th fill's probability under different model assumptions. At present such uncertainties must be encoded through extended suffixes in the YODA `AnalysisObject` paths, but a more elegant scheme would generalise the `DbnStorage` classes to be implicitly multi-weighted, without the overheads of maintaining parallel copies of entire histograms (and performing repeated binning lookups). Another natural area for extension is to support an unbinned multidimensional data type, analogous to the "ntuple" types commonly used in particle physics and many other data-science applications. Finally, we note that the template meta-programming techniques used to implement YODA 2 are increasingly embraced and enhanced within the C++ language, and we anticipate further growth of the power, expressiveness, and performance of the YODA system along with future language evolutions.

# 9 Acknowledgements

The authors thank the Marie Sklodowska-Curie Innovative Training Network MCnetITN3 (grant agreement no. 722104) for funding and providing the scope for discussion and collaboration toward this work. AB and CG acknowledge funding via the STFC experimental Consolidated Grants programme (grant numbers ST/S000887/1 & ST/W000520/1 and ST/S000666/1) & ST/W00058X/1), and the SWIFT-HEP project (grant numbers ST/V002562/1 and ST/V002627/1). MF, NR and ST thank Google and the HEP Software Foundation for funding via the 2020 and 2021 Google Summer of Code programmes. YY thanks the Spreadbury Fund and the UCL Impact scheme for PhD studentship funding. JY acknowledges an STFC doctoral studentship via the ScotDIST Centre for Doctoral Training in Data-Intensive Science.

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
