# Peer review of "Consistent, multidimensional differential histogramming and summary statistics with YODA 2"

_SciPost Physics Codebases, doi:SciPost Phys. Codebases 45 (2025) , SciPost Phys. Codebases 45-r2.0 (2025)_

## Round 1 · Referee Report · Giordon Stark (Referee 1) · 2024-4-2

Strengths

  1. This is a well-written paper describing the evolution of a new version of the YODA package which is directly depended on by Rivet.
  2. There is a strong description of the statistical machinery in the beginning which helps ground the work presented by the authors to the mathematics underlying histograms.
  3. It's really nice to have a section highlighting "experiences learned" from previous mistakes. It's very important that this portion makes it through the editorial phase and stays.

Weaknesses

  1. 5.6 is rather short and bereft. Part of the authors claims here are that the usability is crucial, and a python interface is going to be what makes a software package usable.
  2. On pages 7/8, the design principles are nice to have, but I feel like the names assigned to each block is a bit too vague and abstract. "Continuous aggregation" for example does not immediately remind me of what that means. The authors would probably find it best to provide counterexamples or situations in which these design principles do not hold in other existing libraries. Matplotlib is a prominent example of not allowing the so-called "continuous integration". Meanwhile, ROOT allows this. So perhaps trying to highlight that you want the usability of matplotlib but the functionality of ROOT, or similar. I think just more care needs to be taken here.
  3. The last two paragraphs in section 3 are unneeded, and quite strongly opinionated. In fact, I think it's unrealistic and unfair to say that "boost.histogram" is less used (without giving context, or numbers or statistics). One could claim that YODA is not used outside of particle physics.

Report

I recommend that the authors submit this instead to SciPost CodeBases (rather than Phys) given the paper is somewhat more focused on the technicalities of the code base, rather than demonstrating a novel physics result.

Requested changes

  1. The authors should discuss the overlap with the existing boost-histogram python bindings, as well as the user-friendly "hist" package: https://hist.readthedocs.io/
  2. I would appreciate a discussion or acknowledge of the Array API (https://data-apis.org/array-api/latest/) and more specifically, the Uniform Histogramming Interface (https://uhi.readthedocs.io/en/latest/). Section 6 provides a different kind of indexing which is a bit different from standard. Incorporating these, I think, will allow YODA to coexist with other similar histogramming libraries and allow for translation between different toolings. It should also allow for improved usage of YODA outside of HEP.
  3. In Section 6, I was expecting some discussion of the underlying storage of the objects (contiguous in memory or not?) and how the authors ensure the histogram lookups are O(1).
  4. Sections 6 and 7 should be merged, and reorganized. I expect discussions of the plaintext YAML format to be in the "serialization" rather than in "data exchange" which isn't clear to me what that means. I also expect the CLI interface to have it's own subsection to highlight some of the nice functionalities that come out of the box, but without making it a technical documentation.
  5. Bottom of page 9 has a code block that should be a listings block with a reference number. This code block is also not really valid as "ao" and "script_generator" are not defined in this scope.

---

## Round 2 · Referee Report · Giordon Stark (Referee 1) · 2024-5-20

Strengths

All existing strengths from the previous report still apply (copied below for posterity)

  1. This is a well-written paper describing the evolution of a new version of the YODA package which is directly depended on by Rivet.
  2. There is a strong description of the statistical machinery in the beginning which helps ground the work presented by the authors to the mathematics underlying histograms.
  3. It's really nice to have a section highlighting "experiences learned" from previous mistakes. It's very important that this portion makes it through the editorial phase and stays.

In addition to the above, the changes the author makes improves the existing weaknesses and as such, there are no specific weaknesses I can identify now.

Weaknesses

none (see strengths)

Report

Please refer to the following criteria for the codebase submission: https://scipost.org/SciPostPhysCodeb/about#criteria . The following criteria, as the current draft+online documentation exists, remain unfulfilled:

  1. Benchmarking tests must be provided. These should be similar to what we see for boost-histogram here: https://www.boost.org/doc/libs/develop/libs/histogram/doc/html/histogram/benchmarks.html . I looked through YODA's existing documentation to understand comparisons computationally, given that "much of HEP still needs histogramming to happen in compiled code".

All other criteria are satisfied. Below are some additional minor comments for the authors to consider, but not all of these comments will be requested changes.

6.1: the authors have considered the Single Responsibility Principle when designing their codebase. What I am curious about (given that this is a paper to highlight YODA1 -> YODA2 evolution) is whether these I/O are backwards-compatible, or are self-consistent in some way. If this comes with a schema evolution, I think this would be the place to add a brief sentence if so.

6.1: one blurb I completely overlooked before, is the consideration of HDF5. I don't know if I understand this, as HDF5/Feather/Parquet/etc are suitable for storing columnar data. But this is not necessarily what YODA is dealing with when serializing. I am not sure I understand what the authors intend here.

6.1: this is more about the long-term adoption of YODA from python. In the last paragraph (this somewhat overlaps with UHI), there are some details about how to access bins/points within a given histogram object. It is somewhat curious to me that one accesses systematic variations via brackets $\texttt{[]}$, while multi-dimensional indices are via the pound sign (or hashtag) "#". While I can imagine the motivations here to split the various access methods, it feels a little more work for a user to remember which is which, especially if they use this from python where the "@" operator is meant to indicate matrix multiplication. If I were to use this package, I would probably be a little confused about the "@" here (as I don't think this is used in ROOT and might really be YODA-specific) and I might want to tentatively expect $\texttt{histo[1,2]}$ to work just as well as $\texttt{histo["systematic", 1,2]}$. As this comment is more about the functionality of the code rather than the paper, I will let the authors take this feedback, but it does not impact the publish-ability of the current draft.

7: I have not checked this, but as pandas does not use numpy (and uses their own internal library), is YODA currently compatible with pandas? Ideally, if one relies on the Array API [and not Numpy's API], then one is compatible. Otherwise, the first paragraph needs to be slightly rephrased as not all popular computational libraries would be compatible (if pandas is not). Similar comment for awkward-array which is increasingly more popular for non-regular structured data, such as handling histograms of arbitrary dimensions. It evokes a larger question about whether YODA is intended to only work with numpy objects, and the expectation is that when using other libraries such as pandas/awkward, we would build numpy-like views of the native array objects in those libraries to pass into YODA as needed, or if YODA should be able to support non-numpy-array objects?

One last comment is that I noticed there is no discussion about GPU computation (supporting CuDA). It is likely, given the current design of YODA and allowing for MPI support, that it becomes rather inefficient to use this package now for GPU. Are there plans to support this avenue, e.g. with a YODA-GPU plugin, or will the authors only maintain YODA for CPU-based computing?

Requested changes

  1. provide a table highlighting the computational benchmarks for various dimensionalities of histograms (filling/access) [required criteria]
  2. clarify the planned integration of HDF5.
  3. clarify if other non-numpy libraries are currently compatible without needing the explicit numpy-like view.
  4. clarify support for GPU, in addition to CPU.

Recommendation

Ask for minor revision

  • validity: top
  • significance: high
  • originality: top
  • clarity: high
  • formatting: perfect
  • grammar: perfect

Author:  Christian Gutschow  on 2024-11-10  [id 4950]

(in reply to Report 1 by Giordon Stark on 2024-05-20)

Thanks again for the inputs. We just respond to the point about benchmarking, but take on board the other comments for later design iterations.

  1. Benchmarking tests must be provided. These should be similar to what we see for boost-histogram here: https://www.boost.org/doc/libs/develop/libs/histogram/doc/html/histogram/benchmarks.html . I looked through YODA's existing documentation to understand comparisons computationally, given that "much of HEP still needs histogramming to happen in compiled code".

Re 1: It is not made fully clear what "benchmarking" means in this context, as performance is (as the Boost Histogram page notes) rarely a performance bottleneck, and the paper does emphasise that the bias in YODA2 is toward a mix of C++, mathematical correctness/consistency, feature set, and user-friendliness. CPU performance tests are certainly not required of all SciPost Codebases papers, as performance in that sense is not always a relevant metric -- arguably histogramming is in that position.

Examples/unit tests are provided in the package, and perhaps already fulfill a definition of behavioural benchmarking, but we have now added timing tests to that set. These cannot be exactly the same as Boost Histogram's as it tests performance in a very simplified mode of integer counting while YODA's histograms also track fractional weights, weighted axis moments, etc. In discussion with Hans Dembinski we were also unable to reproduce the setup used to calculate clock-cycle counts, so have chosen not to compare timings one-for-one. We have, however, added a paragraph on the results of testing, showing adequate performance for most/all purposes and good scaling both with dimensionality and with binning metric.

Re 2: An alternative HDF5 format is planned to complement the existing ASCII format. The relevant paragraph in the I/O section seems clear enough to us.

Re 3: Computational Python libraries other than numpy are currently not supported.

Re 4: No support for GPU is planned, seeing as histogramming is currently not a critical bottleneck for anyone as far as we can tell.

---

## Round 2 · Author Response

Thanks for the close reading, and we're glad that the explanations of the maths foundations and the design lessons learned were appreciated. We have updated the paper in response to the comments, and thanks for helping us make it clearer.

However, we disagree on some comments that particularly view the Python interface and the Python sci-comp ecosystem of packages and standards as the relevant context, and explain our justifications for not implementing some requests below.

WEAKNESSES

  1. 5.6 is rather short and bereft. Part of the authors claims here are that the usability is crucial, and a python interface is going to be what makes a software package usable.

The final statement seems a personal opinion. Much of HEP still needs histogramming to happen in compiled code, and YODA is firstly a C++ library for use in applications where C++ is user-facing, and usability is a goal. As the C++ interface is fairly clean, a fairly direct mapping of that to Python is applied and there is relatively little to be said about it.

Our previous experience was that it was best for user comprehension to make the Python wrapper as thin as possible, rather than implement lots of Pythonisations that effectively define a new API in Python and require users to maintain two mental models of how YODA objects are to be used: a comment to this effect has been added.

  1. On pages 7/8, the design principles are nice to have, but I feel like the names assigned to each block is a bit too vague and abstract. "Continuous aggregation" for example does not immediately remind me of what that means. The authors would probably find it best to provide counterexamples or situations in which these design principles do not hold in other existing libraries. Matplotlib is a prominent example of not allowing the so-called "continuous integration". Meanwhile, ROOT allows this. So perhaps trying to highlight that you want the usability of matplotlib but the functionality of ROOT, or similar. I think just more care needs to be taken here.

Matplotlib is not a histogramming package, but a plotting one: the exception is its thin wrapping of numpy.histogram to mimic Matlab's historic interface. This is the reason for Matlab and numpy already being the cited counterexamples.

ROOT as the most prominent (though not the original) exemplar of the continuous-aggregation mode is already discussed after the bullet list, and we've tweaked the wording in the definition-list a little. Note that continuous aggregation is unrelated to the continuous integration mentioned -- we assume this was just a typo.

  1. The last two paragraphs in section 3 are unneeded, and quite strongly opinionated. In fact, I think it's unrealistic and unfair to say that "boost.histogram" is less used (without giving context, or numbers or statistics). One could claim that YODA is not used outside of particle physics.

We have removed reference to Boost Histogram usage, as it's not particularly relevant and it's hard to evidence use as literature searches for utility packages just find the authors' writeups.

But we disagree that the remainder is opinionated: as the footnote highlights, the intention is to spell out the ecosystem and the reason for yet another histogramming package (although most of these decisions were made ~15 years ago with YODA v1!) rather than to criticise. We are not aware of any unfactual statements made here about the comparable packages, and consider this section highly relevant to potential users trying to weigh the pros and cons of different solutions.

REQUESTS

  1. The authors should discuss the overlap with the existing boost-histogram python bindings, as well as the user-friendly "hist" package: https://hist.readthedocs.io/

Again, the focus on Python does not seem appropriate: YODA is a user-friendly C++ interface, which is then semi-trivially mapped into Python. We spend almost the entirety of the paper talking about the C++ design and implementation, and so the relevant primary comparators are to other C++ interfaces.

The boost-histogram C++ and Python bindings expose the same API which implements general histogramming but without the user-facing syntactic sugar, and Hist implements some UI refinements but a) only in Python, and b) still rather verbose wrt the HEP baseline of ROOT's API, and requiring a fairly high level of Python stdlib fluency for some operations (e.g. functools.reduce(operator.mul, ...) used in its quickstart example). We think it is not a useful thing to spend time critiquing other packages' APIs in a different language, though.

  1. I would appreciate a discussion or acknowledge of the Array API (https://data-apis.org/array-api/latest/) and more specifically, the Uniform Histogramming Interface (https://uhi.readthedocs.io/en/latest/). Section 6 provides a different kind of indexing which is a bit different from standard. Incorporating these, I think, will allow YODA to coexist with other similar histogramming libraries and allow for translation between different toolings. It should also allow for improved usage of YODA outside of HEP.

Again these are Python-specific: technical capabilities as well as ecosystem differences mean that such standards can't just be mapped across languages, though aspects of our API design were influenced by Python native capabilities.

Interface definitions like UHI are even more localised: it seems to have been developed and deployed entirely within the Python SciKit-HEP community, and is only "universal" within the scopes envisaged for those tools. From our reading, its plotting interface does not envisage plotting of asymmetric or systematic uncertainties, nor representing overflows or irregular tilings, and mixes up statistical moments with display choices (e.g. in prescribing what error bars in profile-type data objects are to mean).

So UHI looks a potentially interesting idea, though limited to Python rendering backends and a subset of functionality, and requiring a separation of value and axis objects which is at odds with YODA's evolved design. It's not clear at present how much value would be added by retro-fitting the "missing" UHI API elements, as the support does seem to be within closely related SciKit-HEP packages, but we will discuss with the UHI author(s).

In total, as there is currently such a mismatch, we feel it would not be appropriate to reference these, only to say that they are not obviously relevant.

  1. In Section 6, I was expecting some discussion of the underlying storage of the objects (contiguous in memory or not?) and how the authors ensure the histogram lookups are O(1).

The text on local/global indices has been improved, and a sentence added to confirm that the actual bin contents are stored contiguously in memory. Lookup of the content from either local or global indices is trivially O(1) given the storage, but this is now explicitly stated. Lookup from coordinates is not, and the strategies used to optimise that index computation are already discussed.

  1. Sections 6 and 7 should be merged, and reorganized. I expect discussions of the plaintext YAML format to be in the "serialization" rather than in "data exchange" which isn't clear to me what that means. I also expect the CLI interface to have it's own subsection to highlight some of the nice functionalities that come out of the box, but without making it a technical documentation.

We strongly disagree that these sections should be merged: data exchange and visualisation are completely different facets, both in principle and in implementation. Also, there seems to be a confusion between in-memory serialisation and file formats here: the two are distinct and hence discussed in distinct subsections, but within the common (and hence generically titled) section on data exchange.

We felt that it was more useful to note cognate CLI abilities where relevant to features, rather than to isolate them into a distinct section: we have highlighted all the CLI features of most interest for this level of document via these contextual notes, and a dedicated subsection would be both disjoint and very short.

  1. Bottom of page 9 has a code block that should be a listings block with a reference number. This code block is also not really valid as "ao" and "script_generator" are not defined in this scope.

We disagree that this should be a listings block: it's an inline example snippet, not something to refer back to. We have now indicated the meanings of the variables in the text, thanks for the note!

REPORT

I recommend that the authors submit this instead to SciPost CodeBases (rather than Phys) given the paper is somewhat more focused on the technicalities of the code base, rather than demonstrating a novel physics result.

The paper is submitted to SciPost Codebases.

---

## Round 2 · List of Changes

1. Added explanations of contiguous storage and global-index role
  2. Added motivation of thin Python wrapper
  3. Removed mention of Boost Histogram usage level, better explained relation to other packages

---

## Round 4 · Author Response

We thank the referee for his careful reading of this release note.
We addressed the feedback in this resubmission (v4 on the arXiv).

---

## Round 4 · List of Changes

1. Replaced the SciPost style template
  2. Added a short summary of timing tests

---

## Editorial Decision

published